# Genomic Association Mapping of Apparent Amylose and Protein Concentration in Milled Rice

**Jasper Benedict B. Alpuerto [1], Stanley Omar P. B. Samonte [2], Darlene L. Sanchez [2],\*, Peyton A. Croaker [2], Ya-Jane Wang [3], Lloyd T. Wilson [2], Eric F. Christensen [2], Rodante E. Tabien [2], Zongbu Yan [2] and Michael J. Thomson [4]**

[1] Bayer Research and Development Services (Bayer Crop Science), 700 Chesterfield Parkway W., Chesterfield, MO 63017, USA; jasper.alpuerto@bayer.com
[2] Texas A&M AgriLife Research Center, 1509 Aggie Drive, Beaumont, TX 77713, USA; stanley.samonte@ag.tamu.edu (S.O.P.B.S.); pacroaker@tamu.edu (P.A.C.); lt.wilson@aesrg.tamu.edu (L.T.W.); eric.christensen@aesrg.tamu.edu (E.F.C.); rodante.tabien@ag.tamu.edu (R.E.T.); zongbu.yan@ag.tamu.edu (Z.Y.)
[3] Department of Food Science, University of Arkansas, Food Science Building, 2650 N. Young Ave., Fayetteville, AR 72704, USA; yjwang@uark.edu
[4] Department of Soil and Crop Sciences, Texas A&M University, 343C Heep Center, College Station, TX 77845, USA; michael.thomson@ag.tamu.edu
\* Correspondence: darlene.sanchez@ag.tamu.edu; Tel.: +1-(409)-245-8609

**Abstract:** Apparent amylose and protein concentrations (AAC and PC) affect the milling, cooking, and eating quality of rice. This study was conducted to assess the phenotypic and genotypic variation and to identify genomic regions and putative candidate genes associated with AAC and PC in milled rice grain. Two hundred and seventeen and 207 diverse rice accessions were grown at the Texas A&M AgriLife Research Center in 2018 and 2019, respectively. Milled rice samples were analyzed for AAC and PC using the iodine colorimetry and Dumas method, respectively. Genome-wide association studies (GWAS) for AAC and PC were conducted using 872,556 single nucleotide polymorphism (SNP) markers following the mixed linear model. Significant variation among the accessions was found for both variables each year. Associations between 32 SNPs with PC and seven SNPs with AAC were detected. Gene models linked to these SNPs have a wide range of biological functions, including protein and carbohydrate metabolism, DNA methylation, and response to abiotic and biotic stresses. Seven of the identified SNPs colocalized with previously reported quantitative trait loci (QTL) for protein concentration. Fine-mapping of significant genomic regions and gene validation are necessary for this information to aid in marker-assisted breeding for improved grain quality.

**Keywords:** rice; amylose; protein; grain quality; genome-wide association study

## 1. Introduction

The development of high-yielding rice with improved grain quality is advantageous for producers, as it is one of the key determinants of cultivar marketability. Rice grain is evaluated by its appearance, eating and cooking quality, and nutritional value. These, in turn, affect the level of consumer preference a cultivar receives [1].

Apparent amylose concentration (AAC), gel consistency, and gelatinization temperature are major factors that affect rice eating and cooking quality [2]. Among these, amylose has been regarded as the most important factor in determining the sensory properties of cooked rice [3,4]. Amylose is one of two components that build starch granules in rice grain. Amylose has a straight-chain structure, while amylopectin has a branched-chain structure [5]. The classifications of rice based on AAC are waxy (1 to 2%), very low (2 to 12%), low (12 to 20%), intermediate (20 to 25%), and high (>25%). AAC affects the physical appearance of milled rice. High-AAC grains are more translucent [6] than low-AAC grains, which appear whiter [7]. In addition, AAC levels affect the texture of cooked rice. Cooked

waxy rice has a sticky texture, while high-AAC rice separates and is fluffy. AAC also affects the hardness of cooked rice, in which low-AAC rice is softer while high-AAC is firmer [8]. High AAC is positively correlated with higher resistance starch, the portion of starch that has similar characteristics to beneficial dietary fiber [9]. Resistant starch is not readily digested by amylases and is affected by the ratio of amylose and amylopectin [10]. Brown and white rice consumption typically result in about 1.7 g and 1.2 g resistant starch per 100 g consumed, respectively. This equates to glycemic indices of 66 for brown rice and 72 for white rice [11]. Consequently, high-AAC rice is considered to have the potential to be classified as low glycemic index food, which is beneficial to consumers with a higher propensity for diabetes [12].

Grain protein concentration (PC) is another factor that affects the eating and cooking quality of rice [1]. The nutritional value of milled rice grain consists primarily of starch and protein. The latter is in lower concentration compared with other cereals and relatively low in most rice cultivars [13]. Rice grain PC can range from 4.9 to 19.3% in indica cultivars and 5.9 to 16.5% in japonica cultivars [14]. Similar to AAC, grain PC affects the digestibility and flavor of cooked rice [15]. It has been reported that rice with low grain PC has a more desirable flavor (i.e., sweet and aromatic) compared with high grain PC [16]. Grain PC in rice comprises different components—primarily glutelins and prolamins—that have different amino acid profiles and affect the nutritional value and overall digestibility of rice grain [13,17]. Thus, producing high-grain-quality rice requires understanding and balancing eating and cooking qualities concerning grain PC.

Depending on the geographical region or culture, rice breeding programs aim for different rice types based on grain quality traits, including AAC and PC. This is due to varying consumer demands or preferences for food products made from or with rice. For example, soft and sticky rice is preferred in east Asian countries such as South Korea and Japan, while fluffy and nonsticky rice is preferred in India and South America [6]. With the increasing emphasis on the importance of health, the quality and nutritional value of cereal crops, including rice, are increasingly considered as the primary objectives of plant breeders [4]. To facilitate improvements in quality and nutritional value, the determination of the genetic basis of traits affecting grain quality is necessary to increase cultivar improvement efficiency. The challenge arises from the relationships among traits, as some may be undesirable. For example, high grain PC can reduce head rice percentage and has been associated with a reduction in cooking and eating quality. Increasing AAC affects grain PC, and both are major components in rice grain. These suggest that selection for improvement of AAC and PC in rice grain should involve a complex understanding of their accumulation and should consider consumer preferences.

Recent advances in gene editing suggest the potential for rice grain AAC and PC improvement. Gene editing technologies such as CRISPR/Cas9 have been shown as effective tools in altering rice AAC [18,19] and PC [20,21]. However, this approach can be costly, continues to challenge geneticists and breeders in introducing stable target genes into cultivars, and consumer acceptance of gene-edited products differs among countries. Thus, the use of naturally occurring genetic variation to improve grain quality is still a favored option [22]. A genome-wide association study (GWAS) is a powerful tool for analyzing genetic relationships between single-nucleotide polymorphisms (SNPs) and phenotypic variance. In addition, GWAS offers greater resolution than linkage mapping due to the higher recombination among random genotypes within the population accumulated during their respective evolution [1]. The GWAS approach has identified previously reported and/or novel loci associated with AAC and PC in diverse rice populations [1,23–25]. The use of GWAS and marker-assisted selection (MAS) could be a cost-effective way to develop rice cultivars with desired levels of grain AAC and PC and may be especially applicable in a breeding population developed from a cross between parents that vary widely in AAC and PC.

In this study, GWAS was performed on a rice population with the objectives being to evaluate the phenotypic and genotypic variation of AAC and PC in milled rice grain, identify associated genomic regions, and develop candidate gene models.

## 2. Materials and Methods

### 2.1. Plant Materials and Field Plots

Diverse rice lines consisting of 217 and 207 accessions were evaluated in 2018 and 2019, respectively. Accessions from the USDA rice minicore collection [26] comprised 62.7% of the population, with the addition of US cultivars (14.5%), elite inbred lines (10.9%), and test hybrids (3.6%) developed at the Texas A&M AgriLife Research in Beaumont, TX, and foreign cultivars and landraces (8.2%). In terms of the rice subspecies composition, 70.9% are *japonica*, 7.7% are *indica*, 0.9% are *aus*, 19.1% are admixed, and 1.4% are unknown. The accessions were planted (drill-seeded) at the Texas A&M AgriLife Research Center in Beaumont, TX. An augmented, randomized, complete block design was used with five check cultivars (Antonio, Cheniere, Cocodrie, Colorado, and Presidio) replicated in four blocks. In both years, each accession was planted in a three-row plot that was 2.4 m-long. Irrigation was the same for both years, in which the fields were flash-flooded after drill-seeding and as needed within one month after planting. Permanent flooding (water depth at 10 cm) was maintained starting at one month after planting until harvest. For 2018, fertilization was split into 108 and 129 kg ha$^{-1}$ N for first and second applications, respectively. For 2019, fertilization was split into 59, 129, and 47 kg ha$^{-1}$ N for the first, second, and third applications, respectively. At maturity, approximately 30 days after heading, a row from each accession was bulk-harvested, threshed, and oven-dried for two days at 38 °C. Paddy rice samples of each accession were dehulled, milled, separated for whole grain, and ground in preparation for grain quality evaluation. A cyclone sample mill from Udy Corp (Fort Collins, CO, USA) with a 0.84-mm mesh size was used to grind milled rice samples to flour.

### 2.2. Grain Apparent Amylose and Grain Protein Determination

One hundred milligrams (0.1 g) of rice flour from each accession was used to estimate AAC using iodine colorimetry following the Juliano method [27]. Iodine color formation was used to measure the absorbance, which was compared against a standard curve. N concentration was estimated for each accession each year from a 0.15 g subsample using a LECO FP-528 Nitrogen/Protein Determinator (St. Joseph, MI, USA), which uses the Dumas combustion method [28], with grain PCs estimated assuming a nitrogen-to-protein conversion ratio of 1:5.95.

### 2.3. Marker Data and Genome-Wide Association Analysis (GWAS)

Leaf samples were collected from all accessions in the 2018 field experiment. The leaf tissues were kept in liquid nitrogen during sample collection and stored in dry ice during transport from Beaumont, TX to College Station, TX. DNA extraction was performed using standard protocol for leaf tissue with the Thermo Fisher Scientific KingFisher Flex (Thermo Fisher Scientific, Waltham, MA, USA). The DNA samples were sent to the Texas A&M AgriLife Genomics and Bioinformatics Service (TxGen) in College Station, TX, USA for skim sequencing with $1\times$ genome coverage, which was performed with their AgSeq library preparation and next-generation sequencing pipeline [29]. Alignment to the *Oryza sativa* subsp. japonica cultivar Nipponbare reference genome sequence, developed from the International Rice Genome Sequencing Project (IRGSP) Build 5 [30,31] was used to identify 1,075,302 SNPs. Imputation was conducted using BEAGLE V4.0 [32] after initial filtering. Low-quality SNP filtering—removing markers that were <2.5% minor allele frequency and >5% missing data—resulted in 872,556 SNPs, which were then used in the GWAS analysis. Linkage disequilibrium (LD) decay was estimated using TASSEL 5.2.61 [33] to define the appropriate resolution for association mapping. The square of the coefficient of determination ($r^2$) between pairs of SNP markers in a sliding window of 50 markers

was calculated with the mean $r^2$ computed every 10,000 base pairs (bp). LD decay was determined as the distance (in bp) wherein the mean $r^2$ dropped to half its maximum value. The TASSEL 5.2.61 function Mixed Linear Model (MLM) with optimum compression level and re-estimation after each marker as variance component estimation was used to identify significant markers. Principal component analysis (PCA) was used to account for population structure, kinship using the centered identity by state (IBS) method [34], and Bonferroni multiple testing correction was applied in genotyping for marker–trait associations. The R package qqman [35] was used to generate Manhattan, PCA, and quantile–quantile (Q-Q) plots.

Candidate gene models that harbor the significant SNPs were identified using the Nipponbare IRGSP Build 5 genome browser in the Rice Annotation Project Database (RAP-DB) (https://rapdb.dna.affrc.go.jp/viewer/gbrowse/build5/, accessed on 5 March 2021) [30,31].

### 2.4. Statistical Analyses

Frequency distributions and analyses of variance (ANOVA) of the phenotypic data, as well as the analysis of allelic effects using a general linear model, were conducted using JMP ver. 14 software (SAS Institute). Multiple testing thresholds for the association analyses were estimated using the "simpleM" [36] statistical program implemented in R. This method accounts for the number of independent tests ($M_{eff\_G}$). Genome-wide significant and suggestive association thresholds were computed by dividing the *p*-value ($\alpha$) = 0.5 (for genome-wide significance threshold) and $\alpha$ = 1 (suggestive significance threshold) with $M_{eff\_G}$. For this study, the multiple testing threshold levels to declare genome-wide significance and suggestive of association were set to $p < 2.83 \times 10^{-7}$ and $2.83 \times 10^{-7} < p < 5.65 \times 10^{-6}$, respectively.

## 3. Results

### 3.1. Grain Protein and Apparent Amylose Concentration

Significant variation among milled rice accessions in both grain PC and AAC was observed during both planting seasons (Table S1). In 2018, PC for the accessions ranged from 4.32 to 9.61%, with a mean PC of 6.21%. Higher PC values were observed in 2019 with an overall mean of 7.61% and a range of 5.08 to 11.81%. In both seasons, most of the rice accessions had PC values ranging from 5 to 9% (Figure 1). There were no significant differences among the check cultivars for PC, with their mean values (averaged across years) ranging from 6.45 (Presidio) to 7.02% (Antonio).

Grain AAC ranged from 0.39 to 33.22% in 2018 and from 0.00 to 26.59% in 2019, with check cultivars ranging from 21.53 (Presidio) to 32.68% (Colorado) in 2018 and from 19.63 (Presidio) to 26.59% (Cheniere) in 2019. Although the sorting order of check cultivars for ACC may have differed slightly between years, the correlation between 2018 and 2019 values was highly significant at r = 0.92 and *p*-value < 0.0001. In both seasons, most of the rice accessions had low to intermediate AAC (Table 1).

### 3.2. Screening for SNP-Trait Associations Using GWAS

The test population in this study exhibited subpopulation structures, with the main clusters being *indica*, *temperate japonica*, and *tropical japonica* subpopulations based on PCA. The first principal component explained 38.9% of the variation and assigned most of the *indicas* and most of the *japonicas* into two groups, while the second principal component explained 12.0% of the variation and separated the *temperate japonicas* and *tropical japonicas* (Figure 2). The top four principal components explained 56.7% of the genetic variation.

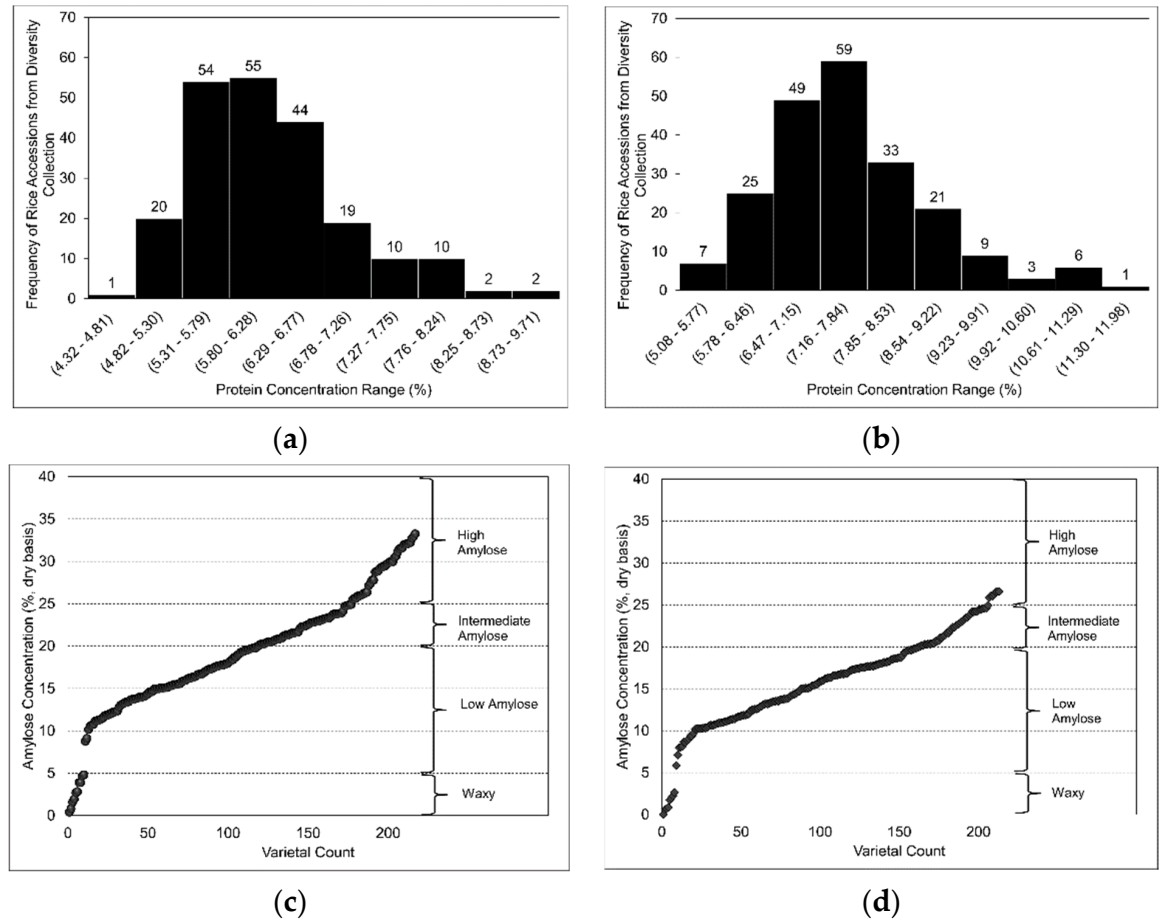

**Figure 1.** Frequency distribution of protein (**a**,**b**) and apparent amylose (**c**,**d**) concentrations of the diverse rice accessions from the 2018 (**a**,**c**) and 2019 (**b**,**d**) seasons.

**Table 1.** Frequency distribution of rice lines based on apparent amylose concentration from phenotyping in Beaumont, Texas in 2018 and 2019.

| Apparent Amylose Classification | Number of Accessions 2018 Season | Number of Accessions 2019 Season |
|---|---|---|
| Waxy | 4 | 6 |
| Very low | 22 | 46 |
| Low | 92 | 105 |
| Intermediate | 59 | 43 |
| High | 40 | 7 |

Thirty-nine significant SNP–trait associations for PC and AAC were found in both years (5% level of significance), with $R^2$ values ranging from 0.104 to 0.259. The Rice Annotation Project Database (RAP-DB) GBrowse function (IRGSP build-5.0) was used to search for candidate genes based on significant SNPs. Candidate gene models were found within 100,000 bp from the significant SNPs location for PC and AAC in both years. This search range was based on genome-wide linkage disequilibrium (LD) decay in this population that was estimated to be ~150,000 bp. This adjustment is within the range estimated in previous findings, where LD decay ranged from close to 100,000 bp to over 200,000 bp [37,38]. The top peak SNP–candidate gene screenings performed for grain PC and AAC in both years are summarized in Figure 3 and Table 2.

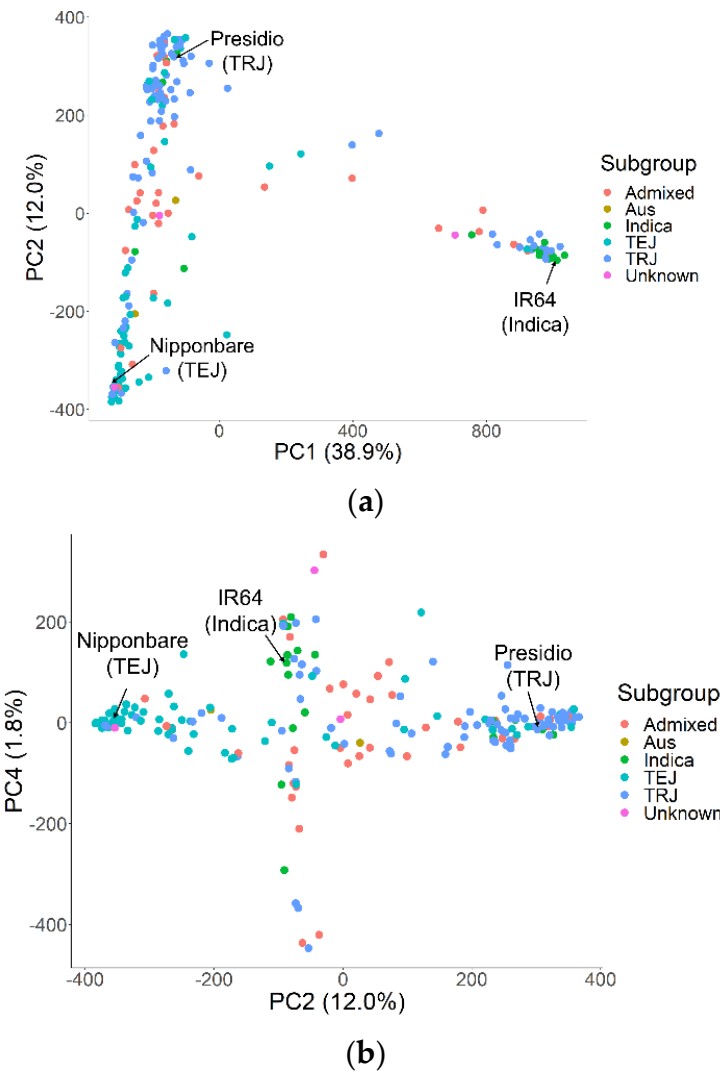

**Figure 2.** Principal component analysis (PCA) of rice accessions used in genome-wide association studies (GWAS) for protein and amylose concentrations, showing (**a**) PC1 by PC2 and (**b**) PC2 by PC4. IR64, Nipponbare, and Presidio are labeled as representatives of the *indica*, *temperate japonica* (TEJ), and *tropical japonica* (TRJ) subgroups, respectively.

In 2018, ten significant SNPs were found based on the genome-wide significance threshold ($p < 2.83 \times 10^{-7}$, after multiple testing) for PC. These SNPs were found in chromosomes 1, 2, 4, 8, and 10. A greater number of significant SNPs for PC were found in 2019 ($p < 2.83 \times 10^{-7}$, after multiple testing). A total of 22 peak SNPs were found with locations spread out among 11 chromosomes, except for chromosome 3, was found. In both years, 7 significant SNPs were found for AAC ($2.83 \times 10^{-7} < p < 5.65 \times 10^{-6}$).

These suggestive significant SNPs were located in chromosomes 4, 7 and 8 in 2018 and in chromosomes 1, 10 and 11 in 2019.

Allelic effects of the top three peak SNPs for PC and AAC in both years are shown in Figure 4. One SNP, S07_25039822, had a significant allelic effect on PC at $p \leq 0.0001$ in 2019. A second peak in chromosome 7, S07_27349671 (not shown) also had a significant allelic effect on PC at $p \leq 0.0001$ in 2019. For AAC, a significant allelic effect was found in SNP S08_22987802 at $p \leq 0.0001$ in 2018, while in 2019, S10_10340782 and S11_14759126 had significant allelic effects at $p \leq 0.05$ and $p \leq 0.01$, respectively.

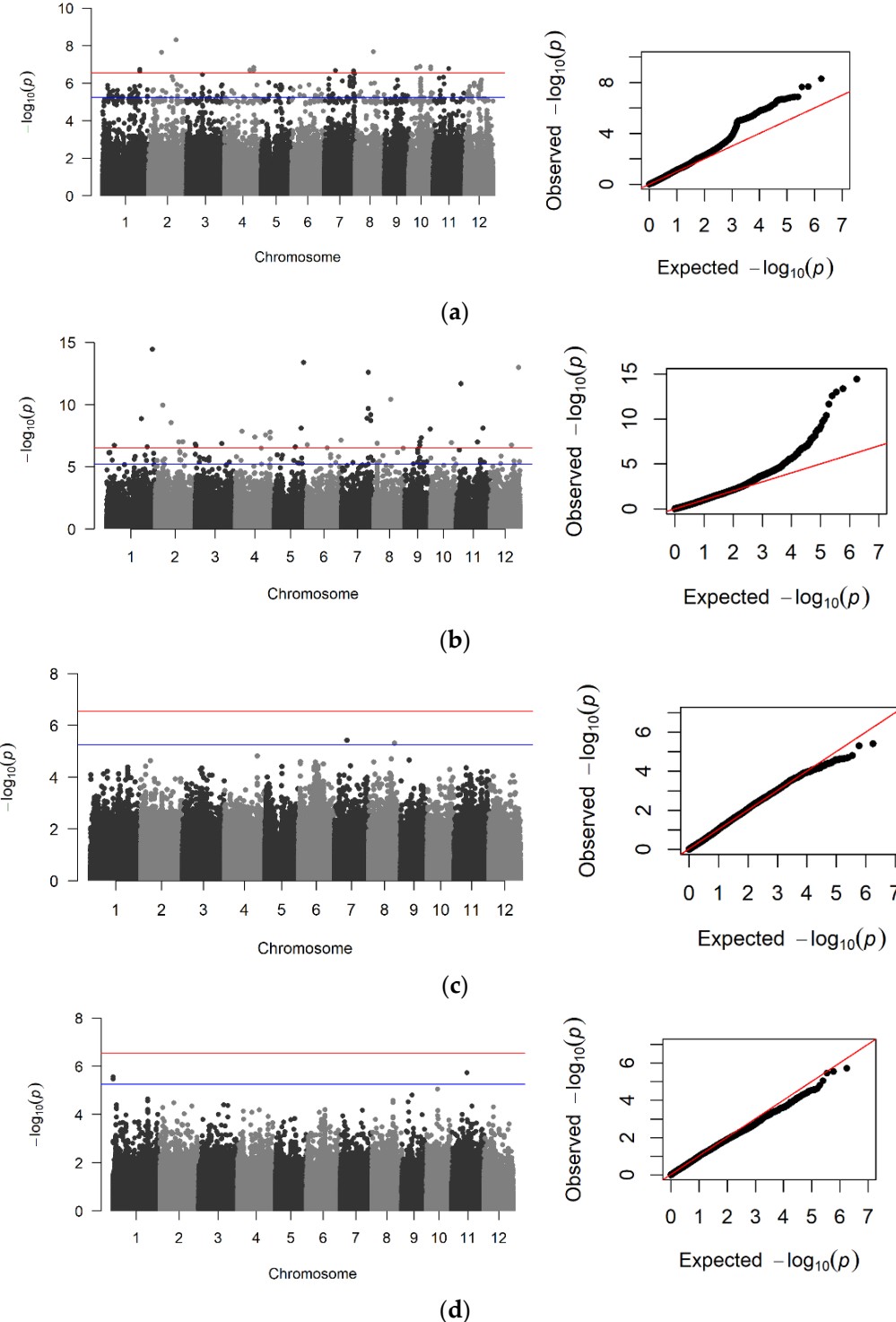

**Figure 3.** Manhattan and quantile–quantile (Q-Q) plots of GWAS for protein concentration (PC) evaluated in 2018 (**a**) and 2019 (**b**) and apparent amylose concentration (AAC) evaluated in 2018 (**c**) and 2019 (**d**). The red line in Manhattan plots represents the genome-wide significance threshold line ($p < 2.83 \times 10^{-7}$), while the blue line in Manhattan plots represents the suggestive significant threshold line ($2.83 \times 10^{-7} < p < 5.65 \times 10^{-6}$) based on the number of independent tests used in the study.

**Table 2.** Significant SNPs detected by GWAS using TASSEL Mixed Linear Model (MLM) associated with milled grain protein and apparent amylose concentrations from the 2018 and 2019 seasons.

| Trait (Season) | SNP Marker | Chromosome | Position (Base Pairs) | *p*-Value | $R^2$ | Allele Effect |
|---|---|---|---|---|---|---|
| Protein Concentration (2018) | S01_36225938 | 1 | 36,225,938 | $1.84 \times 10^{-7}$ | 0.137 | −0.059 |
| | S02_12644707 | 2 | 12,644,707 | $2.23 \times 10^{-8}$ | 0.153 | −0.168 |
| | S02_26708951 | 2 | 26,708,951 | $4.90 \times 10^{-9}$ | 0.167 | −0.167 |
| | S04_28680892 | 4 | 28,680,892 | $1.43 \times 10^{-7}$ | 0.138 | 0.161 |
| | S08_17703647 | 8 | 17,703,647 | $2.12 \times 10^{-8}$ | 0.153 | −0.209 |
| | S10_7561107 | 10 | 7,561,107 | $1.56 \times 10^{-7}$ | 0.137 | 0.077 |
| | S10_10978682 | 10 | 10,978,682 | $1.29 \times 10^{-7}$ | 0.140 | 0.051 |
| | S10_21407969 | 10 | 21,407,969 | $1.73 \times 10^{-7}$ | 0.137 | 0.087 |
| | S10_21408016 | 10 | 21,408,016 | $1.32 \times 10^{-7}$ | 0.138 | −0.225 |
| | S11_15568406 | 11 | 15,568,406 | $1.66 \times 10^{-7}$ | 0.137 | 0.111 |
| Protein Concentration (2019) | S01_32198354 | 1 | 32,198,354 | $1.35 \times 10^{-9}$ | 0.185 | −0.050 |
| | S02_6959467 | 2 | 6,959,467 | $1.09 \times 10^{-10}$ | 0.203 | 0.632 |
| | S02_14624561 | 2 | 14,624,561 | $2.77 \times 10^{-9}$ | 0.177 | −1.330 |
| | S02_25383967 | 2 | 25,383,967 | $9.21 \times 10^{-8}$ | 0.150 | 0.277 |
| | S04_32077706 | 4 | 32,077,706 | $1.61 \times 10^{-8}$ | 0.165 | 0.661 |
| | S04_32077761 | 4 | 32,077,761 | $1.61 \times 10^{-8}$ | 0.165 | −0.661 |
| | S04_32077827 | 4 | 32,077,827 | $4.67 \times 10^{-8}$ | 0.156 | 0.648 |
| | S04_27933097 | 4 | 27,933,097 | $2.76 \times 10^{-8}$ | 0.165 | 0.239 |
| | S04_17895938 | 4 | 17,895,938 | $3.96 \times 10^{-8}$ | 0.154 | 0.399 |
| | S05_27184717 | 5 | 27,184,717 | $4.13 \times 10^{-14}$ | 0.258 | −0.366 |
| | S06_31841867 | 6 | 31,841,867 | $6.94 \times 10^{-8}$ | 0.148 | 0.012 |
| | S07_23761720 | 7 | 23,761,720 | $1.27 \times 10^{-9}$ | 0.184 | −1.020 |
| | S07_25039822 | 7 | 25,039,822 | $2.54 \times 10^{-13}$ | 0.247 | −0.717 |
| | S07_25039804 | 7 | 25,039,804 | $1.99 \times 10^{-10}$ | 0.197 | 0.513 |
| | S07_27349671 | 7 | 27,349,671 | $6.45 \times 10^{-10}$ | 0.188 | 0.841 |
| | S07_27345896 | 7 | 27,345,896 | $1.83 \times 10^{-9}$ | 0.180 | −0.805 |
| | S08_15381100 | 8 | 15,381,100 | $3.76 \times 10^{-11}$ | 0.201 | 1.688 |
| | S09_15114981 | 9 | 15,114,981 | $4.50 \times 10^{-8}$ | 0.157 | 0.229 |
| | S09_23506233 | 9 | 23,506,233 | $9.35 \times 10^{-9}$ | 0.163 | −0.612 |
| | S11_4473024 | 11 | 4,473,024 | $2.07 \times 10^{-12}$ | 0.239 | −0.085 |
| | S11_24669964 | 11 | 24,669,964 | $7.91 \times 10^{-9}$ | 0.168 | −0.079 |
| | S12_26658863 | 12 | 26,658,863 | $9.98 \times 10^{-14}$ | 0.259 | 0.439 |
| Apparent Amylose Concentration (2018) | S04_29329808 | 4 | 29,329,808 | $1.52 \times 10^{-5}$ | 0.104 | 6.726 |
| | S07_11216782 | 7 | 11,216,782 | $3.76 \times 10^{-6}$ | 0.123 | 0.716 |
| | S08_22987802 | 8 | 22,987,802 | $4.80 \times 10^{-6}$ | 0.120 | 7.388 |
| Apparent Amylose Concentration (2019) | S01_253310 | 1 | 253,310 | $2.78 \times 10^{-6}$ | 0.130 | 0.556 |
| | S01_253309 | 1 | 253,309 | $3.39 \times 10^{-6}$ | 0.129 | −0.552 |
| | S10_10340782 | 10 | 10,340,782 | $8.90 \times 10^{-6}$ | 0.122 | 1.646 |
| | S11_14759126 | 11 | 14,759,126 | $1.83 \times 10^{-6}$ | 0.135 | 2.835 |

Due to the observed number of significant peaks, a high likelihood of false positives was suspected. Thus, these peaks were screened for their potential gene models using the GBrowse function of RAP-DB (build 5.0). The putative candidate genes are shown in Table S2. The peak SNPs associated with PC had gene models for carbohydrate and protein regulatory functions (carbohydrate kinase, carbohydrate metabolic process, cysteine-type peptidase activity, protein transporter activity). In addition, gene models for stress response (e.g., response to salt stress, plant disease resistance response protein, response to oxidative stress) were found. Peak SNPs in 2019 were in proximity to gene models for stress response such as disease resistance protein domain-containing protein, pathogenesis-related transcriptional factor and ERF domain-containing protein, oxidative stress response, cytokinin signaling and stress response, and defense responses to bacteria and fungi. Similar to the results of SNP-candidate screening for PC, gene models for peak SNPs for AAC were associated with carbohydrate (similar to alpha-1,2-fucosidase, carbohydrate metabolic process, similar to glutamate decarboxylase, glutamate metabolic process) and protein metabolism (protein phosphorylation and protein serine/threonine kinase activity). Moreover, gene models for the peak AAC SNPs have functions in stress response (such as resistance to rice blast disease and superoxide metabolic process) and enzyme regulation (such as mod-

ulation of the ABA signaling pathway and ABA biosynthesis, regulation of chlorophyll content, enoyl-CoA hydratase/isomerase, and regulation of grain number and yield) that affect growth and development.

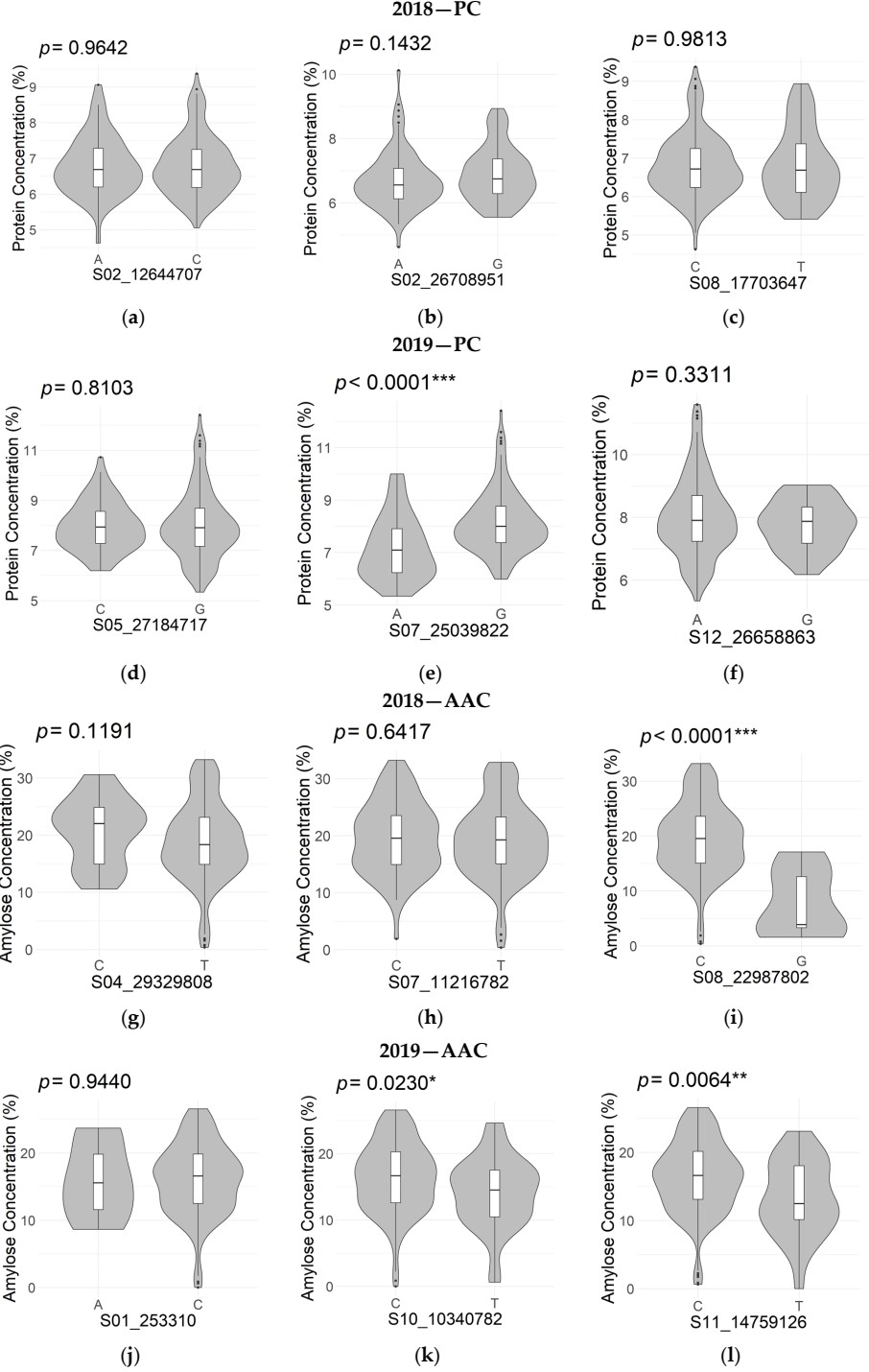

**Figure 4.** The allelic effect of the top three SNPs for protein concentration (PC) in 2018 (**a**–**c**) and 2019 (**d**–**f**) and the top three SNPs for apparent amylose concentration (AAC) in 2018 (**g**–**i**) and 2019 (**j**–**l**) are shown by boxplots within violin plots. The boxplot represents the interquartile range. The central line represents the median value. The gray shape on each side of the boxplot represents all measured data points, and the thickness represents the probability density of the data. The *p*-values of the two allelic effects on each SNP for PC and AAC are shown above each small plot. * Significant at $p \leq 0.05$; ** significant at $p \leq 0.01$; *** significant at $p \leq 0.001$.

SNPs significantly associated with protein concentration colocalized with previously reported quantitative trait loci (QTL) for protein concentration. Two SNPs detected in this study, S06_31841867 and S10_10978682, are within QTL regions reported by Leng et al. [39], namely, *qPC6-h* and *qPC10*, respectively. The SNP S04_17895938 is located within *qRPC-4*, detected by Hu et al. [40]. Four SNPs in chromosome 7—S07_25039804, S07_25039822, S07_27345896, and S07_27349671—are within *qCP-7*, a QTL affecting crude protein concentration in rice reported by Zhang et al. [41].

## 4. Discussion

Improving the grain quality of new rice cultivars has been the focus of breeding programs for several decades [1,2]. Rice grain quality is affected by consumer preferences and depends on how rice is processed or consumed [6]. Apparent amylose and protein concentrations both greatly affect rice grain quality. Most factors that affect grain quality are quantitative traits that are controlled by multiple genes, making it challenging and time-consuming to develop high-quality cultivars. In this study, GWAS was used to characterize phenotypic variation in milled rice protein and apparent amylose concentration to better understand and associate candidate genes.

Wide variations for PC and AAC among rice accessions in both seasons are consistent with previous studies [1,23–25]. Variations in the ranges of PC and AAC between the two seasons are likewise similar to those observed by Wang et al. [1]. The SNPs significantly associated with AAC and PC were different in both years, which suggests that PC and AAC are highly influenced not only by genotype (G) but also by the environment (E) and interaction of the two (G x E). Previous studies have reported that variations in PC and AAC are significantly affected by G, E, and G x E [14,42–46].

This study detected SNPs that are within genomic regions of previously reported QTLs associated with protein concentration [39–41]. The SNP S06_31841867 is within the QTL region *qPC6-h*, which is flanked by markers RM340-RM494 in chromosome 6 [39]. This SNP is associated with Os06g0727200, a gene model with functions related to oxidative stress response. QTL *qPC10*, flanked by the markers RM216-RM467 in chromosome 10 [39], harbors SNP S10_10978682. Wang et al. [1] also detected a peak in chromosome 10 within *qPC10* using GWAS. The SNP S04_17895938 is located within the QTL *qRPC-4*, flanked by markers C22-RG449d [40]. Four SNPs in chromosome 7—S07_25039804, S07_25039822, S07_27345896, and S07_27349671—have shown significant allelic effects ($p < 0.001$) on protein concentration. These SNPs are within a QTL region associated with crude protein concentration, *qCP-7*, flanked by markers R1245-R1789 [41]. The SNP S07_27349671 is in LD with the gene model Os07g0640200, which functions as a carbohydrate kinase, while the SNP S07_25039822 is associated with Os07g0597200, a gene model for serine–threonine protein kinase. The colocalization of the 7 SNPs from this study and known QTL for PC show that the MLM GWAS was effective in minimizing false positives.

Among the 39 significant SNP associations spread across 11 of the 12 chromosomes, gene models associated with grain PC included biological functions linked with DNA methylation, stress (biotic and abiotic) response, hormone regulation, carbohydrate metabolism, and plant growth regulation (Table S1). In chromosome 1, PC was significantly associated with SNPs S01_36225938 and S01_32198354, which harbors gene models with roles in DNA polymerization (Os01g0730900), DNA methylation, response to salt stress (Os01g0811300), transcriptional activation of GA-dependent α-amylase expression, regulation of nutrient mobilization during germination (Os01g0812000), and trehalose biosynthetic process (Os01g0730300). In chromosome 2, gene models for modulation of gibberellin signaling pathway and regulation of plant growth (Os2g0643200) together with protein phosphatases (Os02g0617600; Os02g0224100) and lysine biosynthesis (Os02g0436400) were found to be significantly associated with SNPs S02_26708951, S02_6959467, S02_25383967, and S02_14624561, respectively. In chromosomes 4, 9, and 11, more gene models associated with grain PC were found with functions in disease response (Os04g0621500; S04_32077706, Os11g0592100; S11_24669964), salt stress response (Os04g0620700; S04_32077761), and

oxidative stress response (Os09g0567300; S09_23506233). Moreover, in chromosome 8, S08_17703647 is associated with Os08g0374800, a gene model for carbohydrate kinase, which mainly functions for carbohydrate metabolism. Gene expression profiles of the candidate genes were determined *in silico* using the RiceXPro database [47]. Eight of these candidate genes (Os01g0729600, Os01g0730100, Os02g0644000, Os04g0547600, Os04g0548400, Os04g0561900, Os09g0385700, and Os12g0621500) were highly expressed in the endosperm. These results suggest a broad range of grain protein functions in a developing seed with roles in physiological development and defense response [13,48]. In addition, it illustrates how there are undiscovered regions in the genome associated with grain protein content regulation that can be uncovered to aid in improving grain protein concentration.

For AAC, although only SNPs suggestive of association were found, the gene models associated with these SNPs had functions related to the gene models for grain PC (Table S1). In chromosome 4, gene models associated with S04_29329808 had functions in disease and oxidative stress response (Os04g0573200). In chromosome 1, gene models for ABA biosynthesis and chlorophyll content regulation (Os01g0104600) were found to be associated with S01_253310. In addition, gene models for carbohydrate metabolism regulators (Os10g0339600) were found to be closely associated with S10_10340782. It has been reported that ABA biosynthesis and carbohydrate metabolism have vital roles in regulating grain filling rate in rice [49]. Furthermore, significant allelic effects on AAC were detected in S10_10340782. Granule-bound starch synthase (GBSS), a known enzyme that regulates starch synthesis in rice grains, is encoded by the *Waxy* (*Wx*) gene (Os06g0133000) found in chromosome 6 [50]. In this study, detection of significantly associated SNPs for this locus was only achieved using the TASSEL general linear model (GLM), but not for MLM. This is possibly due to the difference in stringency between the two models, genome coverage used, and AAC variation observed among accessions in both seasons.

## 5. Conclusions

The results of this study added evidence showing the complexity of grain protein and apparent amylose concentration regulation. Thirty-nine SNPs associated with milled grain PC and AAC were found on multiple chromosomes, seven of which were within previously-mapped QTL for PC in chromosomes 4, 6, 7, and 10. Novel loci with candidate gene functions that can be implicated in AAC and PC were also detected. This study increases the understanding of variation in grain PC and AAC in a diverse rice population and identifies DNA markers that can be developed to improve the efficiency with which improved grain quality traits are selected. Functional characterization of the novel candidate genes through fine-mapping, expression analyses, or gene editing is necessary to identify and validate specific genes that can be targeted for improving grain quality.

**Supplementary Materials:** The following supporting information can be downloaded at: https: //www.mdpi.com/article/10.3390/agronomy12040857/s1. Table S1: Amylose and protein concentration of diverse rice accessions grown in Texas A&M AgriLife Research at Beaumont in 2018 and 2019. Table S2: List of gene models located within 100 kilobase pairs of SNPs significantly associated with milled grain protein and apparent amylose concentrations from the 2018 and 2019 seasons.

**Author Contributions:** S.O.P.B.S. conceptualized the study. J.B.B.A. and S.O.P.B.S. designed the experiments and wrote the original draft. J.B.B.A. and D.L.S. conducted the data analyses. S.O.P.B.S. and L.T.W. acquired the funding for this project. Y.-J.W. provided the resources and analyses for amylose concentration, E.F.C. conducted the nitrogen analyses for protein concentration estimation. R.E.T. and Z.Y. provided breeding lines that were used in the mapping population. M.J.T. was instrumental in providing the genotypic marker data. D.L.S., P.A.C., Y.-J.W., L.T.W. and M.J.T. edited the manuscript. All authors have read and agreed to the published version of the manuscript.

**Funding:** The authors appreciate the funding provided by the Texas Rice Research Foundation (Funding Numbers 124214-94410 and 114214-94410) and Texas A&M AgriLife Research (Funding Number 203463-94410).

**Institutional Review Board Statement:** Not applicable.

**Informed Consent Statement:** Not applicable.

**Data Availability Statement:** All the data supporting the results of this article are provided within the article or in the supplementary information. The genotyping data used in this study are part of a dataset deposited at Dryad (https://doi.org/10.5061/dryad.4qrfj6qbs, accessed on 24 March 2021).

**Acknowledgments:** Chersty Harper, Leon Holgate and Leanna Martin are acknowledged for their technical assistance in conducting this study.

**Conflicts of Interest:** The authors declare no conflict of interest.

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
