# Peer review of "Genomic Association Mapping of Apparent Amylose and Protein Concentration in Milled Rice"

_agronomy, doi:10.3390/agronomy12040857_

Round 1

Reviewer 1 Report

A genome-wide associated approach evaluates the roles of SNPs in the regulation of protein and apparent amylose concentration in rice cultivars of diverse origins. The association of grain protein content is linked to the qCP-7 QTL responsible for GPC localization in rice.

major problem

1) No information is provided regarding the percentage degree of milling of rice.

For example, only iodine value can not be suitable for prediction of QTLs related to starch composition. Either DSC-TGA or pasting or rheological properties of milled flour from different rice cultivars must be evaluated, so that the role of starch branching enzyme, debranching enzyme, and granule bound starch synthase enzyme must be correlated to SNPs. which is not carried out.

Frequency distribution data is not acceptable; the entire data should be provided for AAC and PC in supplementry files. 

AAC using iodine value was estimated. However, data for lambda max and blue values is missing, which is crucial for the estimation of AAC in rice. 

So many advancements are available now for the identification of candidate genes associated with major grain quality-related traits. For example:

"A novel major QTL qACE9 for the area of chalky endosperm (ACE) was detected in Hainan and Hangzhou, both mapped in the overlapping region on chromosome 9. It was further fine mapped to an interval of 22 kb between two insertion-deletion (InDel) markers IND9-4 and IND9-5 using a BC4F2 population. Gene prediction analysis identified five putative genes, among which only one gene (OsAPS1), whose product involved in starch synthesis, was detected two nucleotide substitutions causing amino acid change between the parents. Significant difference was found in apparent amylose content (AAC) between NILqACE9 and 9311. And starch granules were round and loosely packed in NILqACE9 compared with 9311 by scanning electron microscopy (SEM) analysis. OsAPS1 was selected as a novel candidate gene for fine-mapped qACE9. The candidate gene not only plays a critical role during starch synthesis in endosperm, but also determines the area of chalky endosperm in rice. Further cloning of the QTL will facilitate the improvement of quality in hybrid rice."

Gao, Y., Liu, C., Li, Y. et al. QTL analysis for chalkiness of rice and fine mapping of a candidate gene for qACE9 . Rice 9, 41 (2016). https://doi.org/10.1186/s12284-016-0114-5

Genetic dissection of rice appearance quality and cooked rice elongation by genome-wide association study, The Crop Journal, Volume 9, Issue 6, December 2021, Pages 1470-1480

"I do not observe any such validation study for a specific gene in the present MS, which is essentially required."

Reviewer 2 Report

In this manuscript authors tried to focus on apparent amylose and protein concentrations, the key determinants factors which affect the rice cultivar marketability (nutritional value, cooking quality and eating as well as appearance). To achieve the mentioned targets, the authors performed a genome-wide association study (GWAS) on two hundred and seventeen accessions in 2018 and 2019. The results of their GWAS analysis showed the selection of 872,556 markers. In addition, the authors stated the significant variation among the accession of both years and reported several SNPs. A wide range of biological functions regulated by gene models has been reported in this paper, as well.

This draft is well written and organised well, however, I have some comments and concerns as follows:

Abstract:

The results of this experiment have been generally written and the authors didn’t specify their results in the abstract. I strongly suggest the author to re-write the abstract and I also advise them to follow the format published by the corresponding author (Dr. Darlene L. Sanchez) of this drat “Phenotypic Characterization and Genome-Wide Association Studies on Agronomic Traits Influencing Rice Grain Yield”. Please also indicate the impact of the current result on agriculture and society. What is the next step after the identification of markers?

Introduction:

The introduction is well written, however, I recommend authors to remove and replace the following paragraph because the CRIPSR technique is a well know technique these days and we, as plant researchers shouldn’t ignore the available techniques. Additionally, authors didn’t try to produce new lines or improve the current accessions, they just tried to identify markers, therefore, there is no link between their work and the genome editing/ gene transformation field.

Recent advances in gene editing suggest the potential for rice grain AAC and PC improvement. However, this approach can be costly, continues to challenge geneticists and breeders in introducing stable target genes into cultivars, and consumer acceptance of gene-edited products differ among countries. Thus, the use of naturally occurring genetic variation is still a favored option [17]. A genome-wide association study (GWAS) is a powerful tool for analyzing genetic relationships between single nucleotide polymorphisms (SNPs) and phenotypic variance. In addition, GWAS offers greater resolution than linkage mapping due to the higher recombination among random genotypes within the population accumulated during their respective evolution [1]. The use of GWAS and marker-assisted selection (MAS) could be a cost-effective way to develop rice cultivars with desired levels of grain AAC and PC and maybe especially applicable in a breeding population developed from a cross between parents that vary widely in AAC and PC.

Material and methods

I just have a question in this section, authors mentioned that they evaluated 217 accessions and 213 during two years, however in the abstract, the authors mentioned just 217 accessions were evaluated. Would you please clarify it for me? thanks

Conclusion

The conclusion is too short and also too general, like abstract. Please clearly identify the results of this effort and conclude it.

Best wishes,

Reviewer 3 Report

The GWAS study in Milled Rice is very interesting research domain since it reported an important group involved in development of high-yielding rice. This paper described particularly two important aspects, the apparent amylose concentration and proteins in rice which are major factors that affect the grain quality. Here some related remarks and suggestions:

  1. The introduction section was well structured and informative. But it will be better if the author add some references concerning recent studies on the loci identified through GWAS in rice (few sentences).
  2. M&M section:

Line 101: In Plant materials and field plots, please add information about plant irrigation used.  

Line 107: please correct “0/9%” with “0.9%”.

Line 121: not clear: “100 milligrams (0.1 g) of rice flour of each accession each sample was estimated for 121 AAC by the iodine colorimetry”

Line 129: Please clarify the way of conservation of Leaf samples when sent to Genomics and Bioinformatics Service.

Line 136: could you precise the exact percentages of the removing allele frequency and the missing data of SNPs.

Please add at the end of M&M section statistical analysis part where you put the sentences of lines 125-127 and lines 148-162.

  1. Results

Line 166: delete space in “with a mean”

Figure 2: the letters a&b should be placed on the top of graph to avoid reading confusion. Same thing is for fig 3.

Line 245: the Manhattan plots are small. It would be better if the author increase the size of figures 3 to have clear genetic variants in each figure (a, b, c&d).

  1. Discussion

Line 276-293: this section is like results description.

Line 294-297: the discussion here is short and it should be developed

Line 300: Add references of the previous reported QTLs associated with protein concentration.

The SNPs detected can contribute to understand the complexity of grain protein and apparent amylose concentration regulation, but to be more informative the author need to add more related references. Only 5 references were used in this section (from 30 to 35).

In general the paper is well structured and gives evidence showing some aspects of AAP and PC regulation.

Round 2

Reviewer 2 Report

I have no comments and I suggest accepting the draft in the current format.

Best wishes for the authors 

Author Response

Dear Sir/Madam:

I appreciate your comments and suggestions in this peer review process. Thank you very much for your help in improving our manuscript.

Sincerely,

Darlene Sanchez